# The Carrot Phytoene Synthase 2 (*DcPSY2*) Promotes Salt Stress Tolerance through a Positive Regulation of Abscisic Acid and Abiotic-Related Genes in *Nicotiana tabacum*

**DOI:** 10.3390/plants12101925

**Published:** 2023-05-09

**Authors:** Orlando Acevedo, Rodrigo A. Contreras, Claudia Stange

**Affiliations:** 1Centro de Biología Molecular Vegetal, Departamento de Biología, Facultad de Ciencias, Universidad de Chile, Las Palmeras 3425, Ñuñoa, Santiago 7750000, Chile; 2Laboratorio de Biología Vegetal e Innovación en Sistemas Agroalimentario, Instituto de Nutrición de los Alimentos (INTA), Universidad de Chile, El Líbano 5524, Macul, Santiago 7750000, Chile; 3Research Unit, Department of R&D, The Not Company SpA (NotCo), Avenida Quilin 3550, Macul, Santiago 7750000, Chile

**Keywords:** *DcPSY2*, phytoene synthase, carrot, carotenoids, *Nicotiana tabacum*, salt stress, abscisic acid

## Abstract

*Background:* Carotenoids, which are secondary metabolites derived from isoprenoids, play a crucial role in photo-protection and photosynthesis, and act as precursors for abscisic acid, a hormone that plays a significant role in plant abiotic stress responses. The biosynthesis of carotenoids in higher plants initiates with the production of phytoene from two geranylgeranyl pyrophosphate molecules. Phytoene synthase (PSY), an essential catalytic enzyme in the process, regulates this crucial step in the pathway. In *Daucus carota* L. (carrot), two *PSY* genes (*DcPSY1* and *DcPSY2*) have been identified but only *DcPSY2* expression is induced by ABA. Here we show that the ectopic expression of *DcPSY2* in *Nicotiana tabacum* L. (tobacco) produces in L3 and L6 a significant increase in total carotenoids and chlorophyll *a*, and a significant increment in phytoene in the T_1_L6 line. Tobacco transgenic T_1_L3 and T_1_L6 lines subjected to chronic NaCl stress showed an increase of between 2 and 3- and 6-fold in survival rate relative to control lines, which correlates directly with an increase in the expression of endogenous carotenogenic and abiotic-related genes, and with ABA levels. *Conclusions:* These results provide evidence of the functionality of *DcPSY2* in conferring salt stress tolerance in transgenic tobacco T_1_L3 and T_1_L6 lines.

## 1. Introduction

Carotenoids, which are synthesized by photosynthetic organisms, bacteria, and fungi, are a class of lipidic isoprenoid compounds that originate from secondary metabolism and are recognized for imparting appealing yellow, orange, and red hues to flowers, fruits, and roots [1]. In plants, carotenoids participate in light harvesting during photosynthesis [2] and photo-protection via energy dissipation and detoxification of reactive oxygen species [3,4]. Additionally, these compounds serve as important precursors to essential apocarotenoids, including hormones like abscisic acid (ABA, Figure 1) and strigolactones [5,6,7,8]. Whereas animals do not produce carotenoids internally, these compounds still hold significant nutritional value for them as they act as a source of vitamin A and potent antioxidants. The absence of carotenoids can result in severe conditions like blindness, xerophthalmia, and premature mortality [9].

In higher plants, carotenoid biosynthesis takes place in plastids via the methylerythritol phosphate (MEP) pathway and begins with the generation of 15-*cis*-phytoene from two molecules of geranylgeranyl pyrophosphate (GGPP), a process catalyzed by phytoene synthase (*PSY*). This step represents both the first committed stage and a crucial point for regulating the flow of carotenoids (Figure 1) [10,11,12,13]. Subsequently, 15-*cis*-phytoene is transformed into di-*cis*-ζ-carotene by two subsequent desaturation reactions, catalyzed by phytoene desaturase (PDS) and 15-*cis*-ζ-carotene isomerase (Z-ISO) and then desaturated by ζ-carotene desaturase (ZDS) and carotene isomerase (CRTISO), producing all-trans-lycopene. Next, all-trans-lycopene is converted to different bicyclical molecules. The enzyme lycopene β-cyclase (LCYB) turns all-trans-lycopene into β-carotene, and both LCYB and lycopene ε-cyclase (LCYE) give raise to α-carotene. β-carotene is the substrate of carotene β-hydroxylase-1 (CβHx) needed to produce zeaxanthin, whereas the hydroxylation of α-carotene by two heme-containing cytochrome P450 enzymes (CYP97A3 and CYP97C1) and CβHx produces the yellowish carotenoid lutein. Zeaxanthin is turned into violaxanthin by the action of zeaxanthin epoxidase (ZEP) (and VDE converts violaxanthin into zeaxanthin). After several enzymatic reactions, ABA is produced in the cytoplasm. The initial stage of ABA biosynthesis, which is a crucial step in the production of this hormone, is catalyzed by NCEDs. This makes NCEDs a bottleneck in the biosynthesis process [14,15,16,17,18,19] (Figure 1). ABA participates in many physiological processes including seed dormancy, regulation of plant growth, senescence and control of stomatal aperture, and in tolerance to abiotic stresses such as cold, drought and salinity [14,15,16,18,20].

*PSY* has been functionally characterized in different plant species, such as *Solanum lycopersicum* (tomato) [14,21], *Arabidopsis thaliana* [22], *Zea mays* (maize) [23], *Cucumis melo* (melon) [15,24], *Narcissus pseudonarcissus* (narcissus) [25] and *Capsicum annuum* (pepper) [26], among other species. Whereas *PSY* is present in a single copy in Arabidopsis, paralogous genes have been identified in several economically important crops, such as tomato, maize, tobacco, *Oryza sativa* (rice), melon and *Manihot esculenta* (cassava) [11,14,27,28,29]. In tomato, rice and maize, three paralogous *PSY*s with different roles have been described [11,29]. In tomato, *SlPSY1* is expressed mainly in fruits, whereas *SlPSY2* is more prevalent in leaves and flowers [14,30,31]. *SlPSY3* is predicted to respond to abiotic stress, like *PSY3* from rice and maize that responds also to ABA treatments [11,29]. On the other hand, *PSY1* and *PSY2* of these cereals are induced by light in leaves [11,29,32]. Therefore, the available evidence indicates that there is a range of functional specificity and diversity among *PSY*s in plants. The expression of *PSY* in the root of Arabidopsis increases carotenoid accumulation in response to osmotic and salt stress, as well as the external application of ABA, according to studies by Meier et al. (2011) and Ruiz-Sola et al. (2014) [33,34]. This indicates that ABA controls the production of its own metabolic precursors.

In carrot, one of the few plant species that synthesize and accumulate carotenoids in the storage root in the absence of light, *DcPSY1* and *DcPSY2* paralogs have been identified [35,36]. Both genes present differential expression patterns during leaf and root development. The expression of *DcPSY1* is specifically triggered in leaves, which supports its function in the synthesis of carotenoids in chloroplasts [37,38]. On the other hand, *DcPSY2* exhibited elevated expression levels during storage root development, which is characterized by a marked rise in the overall carotenoid levels [37,38]. Moreover, the expression of *DcPSY2* correlates better than *DcPSY1* with the accumulation of carotenoids in roots of the orange cultivar with respect to wild white (Ws) carrot inbred lines [39]. Arias et al. (2021) found that overexpression of *DcPSY2* in tomato and, transiently, in *Malus domestica* (apple) fruits resulted in a significant increase in total carotenoids, including β-carotene, which was consistent with the expected subcellular localization of *DcPSY2*:GFP in the plastids.

The results obtained from this study suggest that *DcPSY2* is a functional gene that holds great potential to enhance the carotenoid content in fruits of various commercial species. Furthermore, it was observed that the expression of *DcPSY2* and *DcPSY1* is significantly upregulated in response to salt stress in both the leaves and roots of carrot seedlings, and this upregulation is positively correlated with an increase in ABA levels (Simpson et al., 2018). However, only *DcPSY2* expression is induced by ABA in carrot roots, due to the presence of ABREs elements in its promoter [40]. To further investigate the potential benefits of *DcPSY2*, we conducted experiments by overexpressing this gene in *Nicotiana tabacum* (tobacco) and assessing its impact on salt stress tolerance. Our findings showed that *DcPSY2* T_1_ transgenic tobacco lines exhibit an increased survival rate after salt treatment, and this effect is directly associated with the upregulation of abiotic-related gene expression and ABA levels in roots. These results complement and expand on previous research in this area.

## 2. Methodology

### 2.1. PSY2 In Silico Analysis

The *DcPSY2* sequence (Appendix A) was aligned with the *NtPSY2* (Appendix A), and *DcPSY1* (DQ192186.1) and *DcPSY2* (DQ192187.1) protein sequences from *D. carota* were aligned using ClustalX 1.8 [41]. Squalene/phytoene synthase, isoprenoid synthase and trans-isoprenyl diphosphate synthases domains were determined by the InterproScan Web-based tool (http://www.ebi.ac.uk/Tools/pfa/iprscan5/ accessed on 31 March 2023) (Figure 2A). An amino acidic alignment between *DcPSY1* and *DcPSY2* (Appendix A) was included, highlighting the two squalene/phytoene synthase signatures. Multiple alignments of *PSY* amino acid sequences *AtPSY* (GenBank accession number: AAA32836.1), *NpPSY* (CAA55391.1), *DkPSY* (ACM44688.1), *ZmPSY* (ACY70935.1), *DcPSY1* and *DcPSY2* showed the conserved functional domains and amino acids.

### 2.2. Vector Construction

For *DcPSY2* expression in *N. tabacum*, the corresponding coding sequence of 1276 bp was amplified from carrot storage root by RT-PCR with Pfu polimerase (Promega, Madison, WI, USA) and *Psy2*F and *Psy2*R primers (Appendix A) and cloned into pCR8^®^/GW/TOPO vector (Invitrogen^®^, Waltham, MA, USA) according to the manufacturer’s indications. Positive clones were selected by PCR, enzymatic digestion and sequencing (Macrogen Corp, Rockville, MD, USA, Appendix A). Subsequently, pCR8/*DcPSY2* positive clones were recombined into the binary vector pGWB2 [42], to generate pGWB2-*DcPSY2*, then introduced into *Agrobacterium tumefaciens* (GV3101 strain) for stable *DcPSY2* expression in tobacco.

### 2.3. Plant-Material- and Agrobacterium-Mediated Transformation

Seeds of *Nicotiana tabacum* cultivar Xanthi NN were surface sterilized, plated in full-strength Murashige and Skoog (MS) medium supplemented with 1% sucrose and solidified with 0.8% agar, then transferred to a growth room used for in vitro cultivation equipped with a 16 h light/8 h dark photoperiod (photon fluence rate of 100 µmol m^−2^ s^−1^) at 21–23 °C for 4 weeks for *Agrobacterium*-mediated transformation [43]. Antibiotic-resistant shoots regenerated in the presence of 5 mg/L BAP and 1 mg/L IBA were excised from initial explants and transferred to hormone-free MS medium containing 50–100 mg/L kanamycin and 300 mg/L cefotaxime. When seedlings reached about 7 cm in height, with proper root development, the obtained T_0_ plants were transferred to plastic pots (20 × 10 cm) containing a mix of soil and vermiculite (2:1) and cultivated in a greenhouse under a 16 h long day photoperiod illuminated with cool white fluorescent light (115 μmol m^−2^ s^−1^) at 20–23 °C. *NptII* (kanamycin resistance gene) and *DcPSY2* amplification were carried out to select T_0_ tobacco transgenic lines (Appendix A). Three T_0_ transgenic lines (L3, L4 and L6) were selected for gene expression and carotenoid composition in 2-month-old transgenic plants. Then, around 200 seeds of L3, L4 and L6 transgenic T_0_ lines were sown in MS-Suc^−^ (Murashige and Skoog, 1962) supplemented with 100 mg/L kanamycin to select heterozygote and homozygote T_1_ lines for further analysis.

### 2.4. RNA Extraction and Quantitative RT-PCR

Total RNA was extracted using a modified method based on [44]. Samples were collected from 17-day-old wild-type carrot seedlings, from carrots cultivated at 4, 8 and 12 weeks in the greenhouse, from leaves of 2-month-old T_0_ tobacco transgenic plants and from entire roots and leaves of 1-month-old T_1_L3, T_1_L4 and T_1_L6 tobacco transgenic lines. Samples were immediately frozen in liquid N_2,_ then stored at −80 °C. Afterwards, tissue samples were homogenized in the presence of liquid N_2_ and one aliquot (50–100 mg of powder) was incubated with 700 µL of extraction buffer containing 2% (*w/v*) CTAB, 2% (*w/v*) PVP40, EDTA 25 mM, NaCl 2M, Tris-HCl (pH 8.0) 100 mM and 0.05% spermidine trihydrochloride in DEPC-treated water. Samples were incubated at 65 °C for 5 min in 1000 rpm agitation. Thereafter, two chloroform:isoamylic alcohol (24:1) extractions were performed. Supernatant was precipitated with 1 volume isopropanol and 0.1 volumes of 3M sodium acetate (pH = 5.2) for 10 min at room temperature. Samples were centrifuged and RNA pellets were washed with 75% ethanol in DEPC-treated water. Subsequently, pellets were dried and resuspended in 30 µL of ultrapure DEPC-treated water and subjected to concentration, purity and integrity analysis. For cDNA synthesis, 1.0 μg of total DNA-free RNA was mixed with 0.6 mM of oligoAP primer (Appendix A) and Improm II reverse transcriptase (Promega^®^, Madison, WI, USA). Quantitative RT-PCR (qRT) experiments were performed as described [45] in a Stratagene Mx3000P thermocycler, using SYBR Green double-strand DNA binding dye. Specific primers were designed to amplify a specific coding sequence from either carotenogenic genes: *DcPSY1*, (DQ192186), *DcPSY2* (DQ192187), *NtPSY1* (JF461341), *NtPSY2* (JX101475), *NtLCYB* (KC484706), or salt-responsive genes: *NtNCED3* (JX101472.1) and *NtOsmotin* (X61679.1). As normalizers, the gene expressions of *RNAr18S* (AJ236016.1) and *NtEF1α* (AF120093.1) were determined (Appendix A). Relative transcript levels of all genes of interest were obtained by introducing fluorescence results in the described equation [46]. Expression analysis was performed by means of three biological replicates and two technical repeats. In the case of T_1_L3, T_1_L4 and T_1_L6 transgenic lines, three pools of three plants each were used as the three biological replicates that were measured twice. In all cases, the reaction specificities were tested with melting gradient dissociation curves. To test for significant differences in gene expression, results were subjected to a two-tailed Student’s *t*-test (*p* < 0.05, confidence interval 95%) according to the General Linear Models option in the statistical software package Graphpad Prism.

### 2.5. Carotenoid and Chlorophyll Extraction and High-Performance Liquid Chromatography (HPLC-DAD)

Pigments from 2-month-old T_0_ *DcPSY2* L3, L4 and L6 lines and control lines transformed with the pGWB2 empty vector (E/V) were obtained according to [45]. Carotenoids were extracted from 100 mg of pooled leaves with 1 mL of hexane:acetone:ethanol (2:1:1 *v/v*), then centrifuged. The upper phase was collected, then dried under a stream of N_2_ and resuspended in 1 mL of acetone. Total carotenoid quantification was carried out in a Shimadzu Spectrophotometer at 474 nm (absorption of carotenoids and chlorophyll), 645 nm (chlorophyll *a*) and 662 nm (chlorophyll *b*). Pigments were separated in a Shimadzu HPLC (LC-10AT) equipped with a diode array detector using an RP-18 Lichrocart 125-4 reverse phase column (Merck^®^, Rahway, NJ, USA), and a mix of acetonitrile:methanol:isopropanol (85:10:5 *v/v*) as a mobile phase. Separation was performed with a 1.5 mL/min flow rate at room temperature under isocratic conditions. Results were analyzed with LCsolutions^®^ package software version number 1.0.0.1. For phytoene measurements, chromatograms at 285 nm were obtained. Carotenoids were identified according to their absorption spectra, retention time and comparison with specific pigment standards, which were corroborated by comparison with the Carotenoids Handbook [47,48]. All operations were carried out in triplicate, on ice and dark conditions, to avoid photodegradation, isomerization and structural changes of carotenoids.

### 2.6. Norflurazon Treatment

*DcPSY2* T_1_L6 transgenic tobacco seedlings were grown in vitro in full-strength MS media without sucrose (MS-Suc^−^) supplemented with 100 mg/L kanamycin for 14 days. Subsequently, half of the seedlings were placed in MS-Suc^-^ media in the presence of 100 µM of the PDS inhibitor Norflurazon (NFZ) for 7 days [49], then frozen in liquid N_2_ and stored at −80 °C until carotenoid extraction and phytoene quantification were achieved. This experiment was carried out with three biological replicates of 15 pooled T_1_L6 transgenic seedlings from control and herbicide treatment conditions.

### 2.7. Salt Stress Tolerance Assay

T_1_ E/V, T_1_L3 and T_1_L6 transgenic lines were aseptically sown on agar plates containing MS-Suc^-^ supplemented with 100 mg/L kanamycin. After 13 days of in vitro cultivation, different groups of antibiotic-resistant seedlings were placed on Petri dishes containing 25 mL of fresh MS-Suc^-^ medium with or without 250 mM NaCl. After 16 days of the respective treatment, the fresh weight of individual lines was determined, before their transfer to soil for cultivation in greenhouse conditions (16 h light/8 h dark photoperiod, photon fluence rate of 100 µmol m^−2^ s^−1^ at 21–23 °C). Following 21 days of recovery, survival rates of each group of plants were calculated. This experiment was carried out in triplicate using between 10 and 30 T_1_ E/V, T_1_L3, T_1_L4 and T_1_L6 transgenic T_1_ E/V, T_1_L3, T_1_L4 and T_1_L6 transgenic seedlings for each treatment.

### 2.8. ABA Quantification

T_1_ E/V, T_1_L3 and T_1_L6 transgenic seedlings cultivated for 21 days in full-strength MS medium supplemented with 1% sucrose were subsequently transferred to hydroponic medium and allowed to grow for an additional 10 days before acute salt treatment. An amount of 200 mg of leaves and root tissue obtained from hydroponically grown T_1_ (n = 3) seedlings subjected to 250 mM NaCl during 0 and 8 h were flash frozen and ground with a mortar and a pestle in the presence of liquid N_2_ and 4 mL of H_2_0 milli-Q. The mixture was shaken at 4° for 20 min and incubated overnight at 4 °C. Afterwards, the samples were centrifuged at 14,000 rpm for 15 min, and the aqueous phase containing ABA was recovered. ABA determination was performed by the HPLC-ESI-MS/MS system (Agilent 1200 series, MS/MS 6410, Agilent Technologies, Santa Clara, CA, USA) following the instructions described in [50]. The mobile phase was composed of 0,1% formic acid. A sample of 20 µL was separated using a Zorbax C18 (4.6 × 150 mm, 5 µm) reverse phase column with a flow rate of 0.3 mL/min at room temperature. The experiment was carried out in triplicate with 3 plants each.

## 3. Results

### 3.1. DcPSY2 In Silico Analysis and Expression during Carrot Development

The *DcPSY1* (1197 bp) and *DcPSY2* (1276 bp) coding sequences were obtained by conventional RT-PCR, cloned into pCR^®^8/GW/TOPO and sequenced (Appendix A). The predicted proteins *DcPSY1* (398 aa) and *DcPSY2* (437 aa) share 64% identity and present the same amino acids at the active site, substrate binding pocket, Mg^2+^ binding site and the squalene/phytoene synthase domains (Appendix A), isoprenoid synthase, and trans-isoprenyl diphosphate synthase domains found in functional *PSY* enzymes (Figure 2A). A multiple alignment including *AtPSY* (AAA32836.1), *NpPSY* (CAA55391.1), *DkPSY* (ACM44688.1), *ZmPSY* (ACY70935.1), *DcPSY1* (DQ192186.1), and *DcPSY2* (DQ192187.1) showed that *DcPSY2* presents the characteristic functional domains of the *PSY* enzymes such as Trans_IPPS-HH, the aspartate-rich regions, residues that cover the active site and the substrate-binding pocket (Appendix A). Although the amino acid sequence is highly similar, the expression profile of both *PSY* paralogs is different in carrot. *DcPSY1* presents a higher level of expression than *DcPSY2* in leaves, especially at 4 and 8 weeks of development, when it presents 60-fold and 6-fold more expression than *DcPSY2* (Figure 2B). On the other hand, in roots, *DcPSY2* presents 12-fold, 37-fold and 56-fold more expression than *DcPSY1* at 4, 8 and 12 weeks of development, respectively (Figure 2B).

### 3.2. Expression of DcPSY2 in N. tabacum Promotes an Increment in Carotenoid Content

Once the correct *DcPSY2* sequence identity was confirmed, it was cloned and pGWB2-*DcPSY2* (35SCaMV:*DcPSY2*) vector was obtained to transform tobacco leaf disc explants. Transformed plants were selected with kanamycin and genotyped by amplifying the transgene *DcPSY2* (Appendix A). Three lines were selected, as they presented the most similar phenotype with E/V plants. Our analysis of expression levels indicates that line L4 presents a 2.5- and 25-fold higher expression level than L3 and L6, respectively (Figure 3A). Remarkably, the total carotenoid content in leaves of two-month-old transgenic T_0_ tobacco lines were higher in lines L3 and L6 than in L4, which indeed presents a significant reduction. Particularly, line L3 had a 44% increase in β-carotene content, a 41% rise in total carotenoids and a 63% increase in α-carotene, whereas L6 had an increase of 57% in β-carotene and total carotenoid and 59% increased lutein content with respect to E/V lines (Figure 3B). On the contrary, L4 did not present any difference in individual carotenoid composition. Interestingly, the chlorophyll level was also altered in transgenic plants. L3 presents an increment in chlorophyll *a* of 35% and L6 has an increase in chlorophyll *b* and chlorophyll *a* of 57% and 60%, respectively, with respect to E/V lines. In contrast to other results, L4 showed a reduction of 47% in both chlorophylls (Figure 3C).

Looking deeper on the *DcPSY2* function, the accumulation of phytoene in the T_1_ generation of L6 was assessed. T_1_ seedlings of L6 were selected because T_0_ L6 presented the highest increment in carotenoids and chlorophylls. For this purpose, T_1_L6 seedlings grown on MS-agar plates for 14 days were transferred to fresh MS-agar containing the herbicide norflurazon (NFZ) which, via phytoene desaturase inhibition, leads to the accumulation of the colorless carotenoid phytoene. To ensure maximum inhibition of phytoene desaturase activity, a high concentration of NFZ was employed (100 µM; [49]). After 7 days of treatment, 35SCaMV:*DcPSY2* T_1_L6 as well as the T_1_E/V control lines showed evident symptoms of bleaching (not shown). Quantitative HPLC analysis of carotenoids of the aerial tissue revealed that T_1_L6 seedlings showed a 40% increase in phytoene content after NFZ treatment with regard to T_1_E/V plants (Figure 3D), thus demonstrating that phytoene is converted to downstream carotenoids in seedlings grown in vitro. To determine the effect on the endogenous carotenoid pathway, an analysis of the expression of tobacco carotenogenic genes was determined. Interestingly, *NtPSY1* and *NtPSY2* expression was impaired in transgenic lines, whilst *NtLCYB* expression was not affected in L3 and L6 or induced in L4 (Figure 3E), suggesting that the increment in carotenoids and chlorophylls is due to the expression of *DcPSY2.* The results presented here add to the previous discovery made by Arias et al. (2021), demonstrating that *DcPSY2* encodes an active enzyme that results in an increase in carotenoids and chlorophylls.

### 3.3. Transgenic Tobacco DcPSY2 Lines Showed Salt Stress Tolerance in a Chronic Salt Stress Assay

The functionality of *DcPSY2* was also evaluated in terms of conferring salt stress tolerance. For this purpose, we selected 13-day-old T_1_E/V, T_1_L3 and T_1_L6 seedlings to be subjected to an in vitro chronic 250 mM NaCl treatment (Figure 4A). After 16 days of salt stress, *DcPSY2*-expressing seedlings presented a significantly higher fresh weight than E/V plants (Figure 4A,B). Furthermore, when considering the control group, T_1_L3 exhibited a comparable fresh weight to T_1_E/V, and T_1_L6 did not exhibit a significant difference in fresh weight when compared to seedlings that were not subjected to stress (Figure 4B). After salt stress treatment, plantlets were transferred to soil and cultivated in the greenhouse, without stress. The survival rate after 21 days of recovery (Figure 4C) showed that T_1_L3 and T_1_L6 transgenic plants had a 40% and 68% survival rate, respectively, which is remarkably greater than that observed for control plants (16%; Figure 4D), but T_1_L3 presents a similar phenotype to T_1_E/V, contrary to T_1_L6. Considering these results, T_1_L3 lines present a better performance under control and saline stress conditions.

### 3.4. DcPSY2 Transgenic Plants Induce a Boost in ABA and Stress-Associated Genes under Acute Salt Treatment

To evaluate if ABA is synthesized when plants are subjected to salt stress, representative T_1_E/V, T_1_L3 and T_1_L6 transgenic seedlings were exposed to 250 mM NaCl for 8 h. We found that T_1_L3 and T_1_L6 presented a 1.6- and 3-fold raise in ABA levels in leaves, respectively, after NaCl treatment, with respect to unstressed plants (C), which is significantly lower than the 4-fold increase observed in T_1_E/V control plants under similar conditions (Figure 5A). However, T_1_L3 and T_1_L6 present 1.3- and 2.5-fold increases of ABA, respectively, in roots, with respect to unstressed plants (C), whereas T_1_E/V transgenic plants produced no significant changes in ABA levels (Figure 5A). This finding implies that the improved salt resistance in *DcPSY2* plants may be linked to a heightened ABA response in the roots at an early stage. To determine whether the increase in ABA in the *DcPSY2*-expressing transgenic plants was associated with an enhanced expression of salt stress-related genes, we evaluated the transcript abundance of genes that encode enzymes related to ABA production such as the 9-*cis*-epoxycarotenoid dioxygenase 3 (*NtNCED3*) and the osmotic stress response gene, *NtOsmotin* [51]. After 2 h of exposure to salt, there was a 15-fold increase in *NtNCED3* transcript levels in T_1_E/V lines in above-ground tissue, which then decreased to lower levels after 4 h (Figure 4B). Notably, transgenic *DcPSY2* lines showed a progressive increase in *NtNCED3* transcript levels at 2 h and 4 h of salt exposure. Particularly, T_1_L3 presented 4.7- to 7.7-fold increases at 2 h and 4 h, respectively; similar to T_1_L6, with 2.2- and 5.3-fold increases in *NtNCED3* transcript abundance at 2 h and 4 h, respectively (Figure 5B). Interestingly, *NtNCED3* presents a higher basal level of expression in T_1_L3 and T_1_L6 than in T_1_E/V plants (it is due to this that the increment in T_1_L3 and T_1_L6 is lower than that in T_1_E/V at 2 h) (Figure 5B). The relative transcript level of *NtNCED3* in roots increased by approximately 49 times in T_1_E/V, but after 4 h of salt treatment, it decreased and showed no significant differences (Figure 5B). In contrast, transgenic *DcPSY2* lines showed a sustained increase even at 4 h of salt treatment, as observed for T_1_L3 (15- and 29.2-fold increases after 2 h and 4 h, respectively) and T_1_L6 (81.6- and 69.7-fold increases after 2 h and 4 h, respectively) (Figure 5B). In the case of *NtOsmotin*, the transcript levels in aerial tissue remained unaltered between 2 h and 4 h of salt treatment in T_1_E/V, as in T_1_L3, but increased about 7 times in T_1_L6 at 4 h of NaCl treatment (Figure 5C). However, most interestingly, and in correlation with ABA levels (Figure 5A), *NtOsmotin* transcript levels increased in roots inT_1_L3 and T_1_L6 after 4 h of salt treatment (1.9- and 2.6-fold increases, respectively), whereas T_1_E/V showed no significant changes after stress application (Figure 5C). It is important to consider that *NtOsmotin* presents an upper basal level of expression in T_1_L3 and T_1_L6, but not in T_1_E/V plants. Collectively, the T_1_L3 and T_1_L6 transgenic lines exhibited increased and prolonged transcript levels of *NtNCED* and *NtOsmotin*, which corresponded with the levels of ABA in both leaves and roots during the exposure-to-salt-stress treatment.

### 3.5. Endogenous NtPSY Gene Expression Is Enhanced in DcPSY2 Transgenic Lines after Salt Stress Treatment

Some *PSY*s are described to be upregulated upon salt stress, acting as an indicator of salt stress exposure and response [11,29]. Hence, we assessed the transcript levels of *NtPSY1* and *NtPSY2* in the above-ground and underground tissues at 0 h, 2 h and 4 h of exposure to 250 mM NaCl. This analysis aimed to explore how the expression of *DcPSY2* affects the regulation of these endogenous genes during salt stress conditions. In leaves, *NtPSY1* transcript levels remained unchanged in T_1_E/V control lines but increased 2.7-fold in T_1_L3 and T_1_L6 at 2 h of salt treatment, and then dropped to normal levels at 4 h (Figure 6A). In roots, *NtPSY1* transcript levels increased 3-fold after 2 h in T_1_E/V, whereas a 3.5 and 4.7-fold increase was registered in T_1_L3 and T_1_L6, respectively. At 4 h of salt treatment, *NtPSY1* transcript levels dropped to no significant amounts with respect to time 0 in T_1_E/V and T_1_L3, in contrast with T_1_L6, which showed a sustained 3.1-fold increase of these transcript levels (Figure 6A). In the case of *NtPSY2*, it peaked at 2 h of salt stress treatment in leaves, then dropped to basal levels at 4 h in T_1_E/V, but showed a permanent induction for both T_1_L3 and T_1_L6 even at 4 h of treatment. For T_1_L3, there was a 3.0-fold induction at 2 and 4 h of salt stress, whilst T_1_L6 showed a 3.7- to 6.3-fold increase in *NtPSY2* relative to transcript levels at 2 h and 4 h of treatment (Figure 6B). In roots, we found a 4.4-fold increase in induction after 2 h of salt stress in T_1_E/V control lines and 5.6- and 3.7-fold increases in T_1_L3 and T_1_L6 under similar conditions. Moreover, for both T_1_E/V and T_1_L3, *NtPSY2* transcript levels decayed to basal values at 4 h but retained a 4.1-fold increase in induction in T_1_L6 (Figure 6B). The findings suggest that salt stress triggers the upregulation of *NtPSY1* and *NtPSY2* genes in both tobacco T_1_E/V transgenic lines and those expressing *DcPSY2*. However, T_1_L3 and T_1_L6 lines exhibited a sustained increase in transcript accumulation, which may contribute to enhancing their salt stress tolerance.

## 4. Discussion

### 4.1. DcPSY2 Boost Carotenoid, Phytoene and Chlorophyll Content in N. tabacum

In silico analysis showed that *DcPSY2* presents the characteristic functional domains of *PSY* enzymes (Figure 2, Appendix A), suggesting its functionality. This was confirmed by the constitutive expression of *DcPSY2,* which led to up to a 1.4-fold increase in total carotenoids and a 1.5-fold increase of β-carotene in transgenic tobacco plants, indicating that *DcPSY2* is functional in this heterologous model. Previously, we obtained a similar-fold induction in *DcPSY2* transgenic tomato fruits, which produce a 1.2- to 1.3-fold increase in total carotenoid and up to a 1.25-fold increase in lycopene [52]. Subcellular localization [52] and phytoene accumulation (Figure 3D) in lines subjected to NFZ treatment is another factor that supports the functionality of *DcPSY2*. Previous studies have indicated that carotenoid accumulation in photosynthetic tissues is strongly dependent on and co-regulated with chlorophyll biosynthesis and accumulation [53]. Consistent with that study, *DcPSY2* tobacco lines also showed greater chlorophyll levels, especially chlorophyll *a* (Figure 3C).

Total carotenoid and chlorophyll *a* content correlated inversely with the relative transcript abundance of *DcPSY2* transgene in each line. To be more specific, the expression levels of *DcPSY2* that are categorized as low (L3) and moderate (L6) induced significant enhancements in carotenoid and chlorophyll content. On the other hand, the highest expression level (L4) either did not cause any changes or caused a decrease in these levels (Figure 3A–C). This phenomenon has been observed in other studies and species [28,54], demonstrating the intimate association between *PSY* expression level and the regulation of carotenoid accumulation. As an explanation for this phenotype, we hypothesize that low or moderate *DcPSY2* expression leads to an increased carotenoid content through a retrograde-crosstalk mechanism with chlorophyll biosynthesis [55], and that an increment of transgene abundance over a threshold level induces an opposite effect on carotenoid accumulation. This observation was previously reported [28] after expression of *NtPSY* genes in tobacco plants. It is important to remember that inserting a foreign gene into a plant genome is always random, and it may also affect other genes, resulting in gene silencing or altered expression. Furthermore, it is also possible that high expression of *NtPSY2* may associate with lower levels of gibberellins and chlorophylls, since it is reported that *PSY* uses GGPP as a substrate, which is also the precursor for the synthesis of chlorophylls as well as gibberellins [40,55,56]

Transcript levels of endogenous *NtPSY1* and *NtPSY2* were significantly reduced in *DcPSY2* transgenic lines and showed unaltered or decreased *NtLCYB* transcript levels (Figure 3E) in accordance with previous studies carried out on the expression of *DcLCYB1* gene in carrot and tobacco [45,55]. In relation to these findings, we hypothesize that the reduction in the endogenous *NtPSY* transcript levels could be linked to a mechanism targeted to diminishing the metabolic flux towards the carotenogenic pathway, in order to maintain other metabolic routes requiring the common precursor, GGPP, such as the chlorophyll pathway (Figure 1). In accordance with this proposal, authors reported that the expression of *DcLCYB1* in tobacco showed an increment in carotenoid, chlorophyll and gibberellin (GA) through positive feedback on the expression of metabolic precursor genes, such as *DXS* and *GGPPS*, among others [55]. In our work, we noticed that T_1_L6 are smaller in size than T_1_L3 and T_1_E/V plants, even in the absence of salt (Figure 4) during the first month of culture, which could be associated with a reduction in gibberellin levels. These results were comparable to those reported by Fray et al., (1995) from expressing the fruit paralog *SlPSY* in tomato, which resulted in a dwarf phenotype that was attributed to a reduction in GA levels and a slight increase in ABA levels due to competition for the GGPP substrate [28,56]. Additionally, the high *NtPSY* transcript abundance in tobacco transgenic plants produced stunted plants with a decrease in total pigment content, along with phytoene accumulation [28]. However, upon reaching adulthood, T_1_L6 *DcPSY2* plants reached a phenotype similar to T_1_L3 and T_1_EVs, suggesting that their development time is only slower. Together, these results indicate that the increased carotenoid and chlorophyll content in *DcPSY2*-expressing lines was a direct consequence of the level in transgene expression.

### 4.2. Constitutive Expression of DcPSY2 Leads to an Increase in Salt Tolerance and ABA Levels in Transgenic Tobacco

Climate change produces water deficit and salinity in soils, impairing growth and production in plants [57]. Plants have developed mechanisms to cope with stressful situations [58]. Among them, plants activate tolerance through ABA-dependent and independent mechanisms [16]. ABA is synthesized through the carotenoid pathway, being *PSY* and NCED key limiting steps in those syntheses. Previous reports indicate that ectopic expression of *PSY* genes promotes salt tolerance in transgenic organisms [34,59,60]. Similarly, we observed that the constitutive expression of *DcPSY2* transgene enhanced salt tolerance, coupled with higher levels of *NtNCED* expression and ABA level, preferably in roots. Salt tolerance in terms of fresh weight and survival rate after the saline treatment was in direct correlation with *DcPSY2* transcript levels. Even though the survival rate of T_1_L6 plants was better, this did not lead to better growth, which means that T_1_L3 plants might have a higher yield also when the plants are older (Figure 4B,C). The boosted fresh-weight performance under saline stress in transgenic *DcPSY2* plants may also be attributable to reduced water loss under salinity conditions [61,62] and to the significant increment in ABA level [34,63]. Salt stress treatment for 5 h induced ABA accumulation in shoots and roots of Arabidopsis wild-type plants that was accompanied by an increment in carotenoids and expression of *PSY* in the root [34]. Previously, we determined that *DcPSY2* and *DcPSY1* are induced by salt stress, but only *DcPSY2* by ABA, in which ABRE transcription factors mediate the expression of *DcPSY2* [40]. These results complement the previous findings, suggesting that *DcPSY2* confers salt stress tolerance when overexpressed in plants. The functional role of *DcPSY1* in this process remains to be determined.

The increase in transcript abundance of *NtNCED3* correlates with ABA levels, considering that the expression of NCED is required for ABA synthesis. Interestingly, *NtNCED* presented 4- to 6-fold higher basal levels of expression in leaves of L3 and L6 with respect to E/V plants, and presented a sustained induction at 2 h and 4 h in leaves and roots after acute NaCl treatment (Figure 5B). Although in E/V, *NtNECED* was induced only at 2 h, it revealed that the acute stress assay was correctly achieved. *Osmotin,* a later gene in the ABA stress response pathway, was induced preferably in roots of transgenic lines at 4 h (Figure 5C), whereas E/V plants did not show any induction, suggesting that in control plants, the response is delayed.

We propose that the expression of *DcPSY2* in tobacco leads to an increment in the *NtNCED*3 expression level, which permits the production of ABA preferably in the root, leading further to the induction of *NtOsmotin* in the root. This is also sustained by the fact that under salt stress, endogenous *NtPSY* is also upregulated (Figure 6) to permit enough metabolite supply for stress tolerance and for ABA synthesis.

## 5. Conclusions

Taken together, these results lead to the conclusion that *DcPSY2* codifies for a functional enzyme that enhances carotenoid and chlorophyll content in transgenic tobacco lines, and promotes salt stress tolerance in tobacco through an enhancement of the ABA level as well as the expression of genes involved in abiotic stress tolerance. The enhanced salt stress tolerance exhibited by *DcPSY2* transgenic plants could serve as a valuable biotechnological strategy for rootstocks that have significant economic importance. This is especially relevant because rootstocks are in direct contact with the saline stress present in the soil.

## Figures and Tables

**Figure 1 plants-12-01925-f001:**
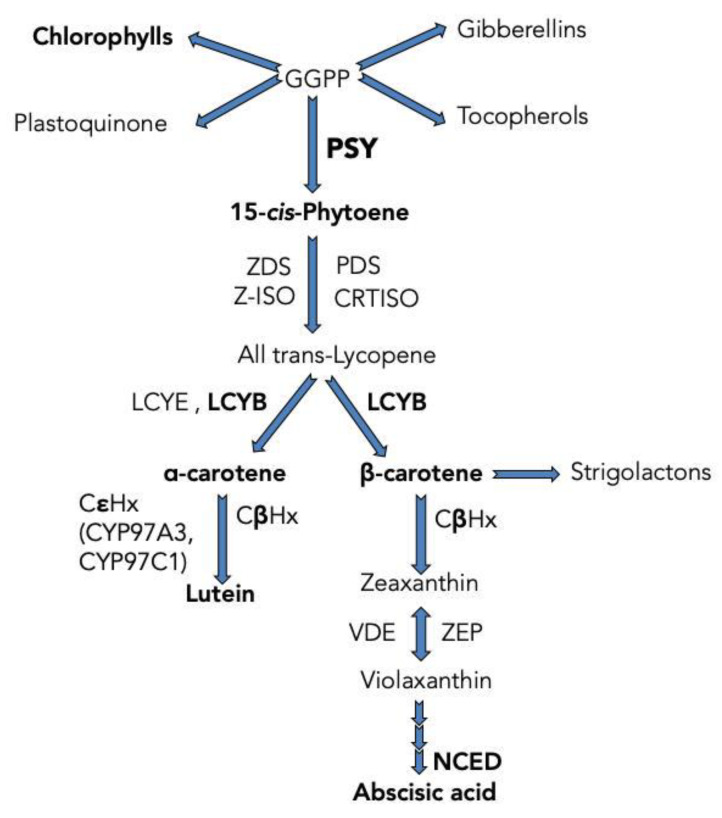
The carotenoid biosynthetic pathway in plants. Two molecules of C_20_ geranylgeranyl pyrophosphate (GGPP) produce 15-*cis*-phytoene, the first carotenoid of the pathway. GGPP is a precursor of Chlorophylls, Gibberellins, Tocopherols and Plastoquinone as well. Abbreviations: phytoene synthase (*PSY*), phytoene desaturase (PDS), ζ-carotene isomerase (Z-ISO), ζ-carotene desaturase (ZDS), carotenoid isomerase (CRTISO), lycopene β-cyclase (LCYB), lycopene ε-cyclase (LCYE), β-carotene hydroxylase (CβHX), ε-carotene hydroxylase (CεHX), zeaxanthin epoxidase (ZEP), violaxanthin de-epoxidase (VDE), 9-*cis*-epoxycarotenoid dioxygenase (NCED).

**Figure 2 plants-12-01925-f002:**
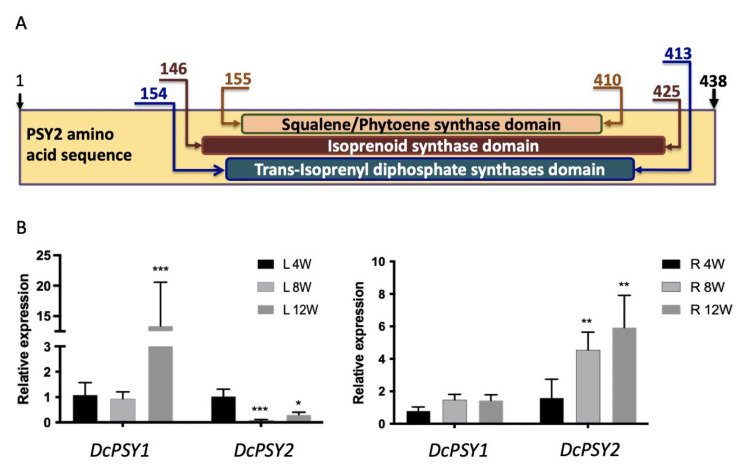
Schematic representation of predicted phytoene synthase domains in *DcPSY2* and expression pattern in carrot. (**A**) The predicted protein has a squalene/phytoene synthase (in green), isoprenoid synthase (in blue) and trans-isoprenyl diphosphate synthase (in brown) domain. The numbers indicate amino acid numbers from the N-terminus of the recently synthesized protein. Protein sequence analysis was performed using InterPro software (http://www.ebi.ac.uk/interpro accessed on 31 March 2023). (**B**) Relative expression of *DcPSY1* and *DcPSY2* in leaves (**left**) and roots (**right**) at 4, 8 and 12 weeks of carrot development. Real time RT-PCR was performed with tree biological replicates measured twice each. *Ubiquitin* gene expression was used as normalizer. As calibrator, *DcPSY1* expression in 4-week-old leaves (L 4W) and *DcPSY1* in 4-week-old roots (R 4W) were used and settled in 1. Error bars represent standard error of the mean (three technical replicates; asterisks indicate Student’s *t*-test significant differences * *p* < 0.05, ** *p* < 0.01, *** *p* < 0.001).

**Figure 3 plants-12-01925-f003:**
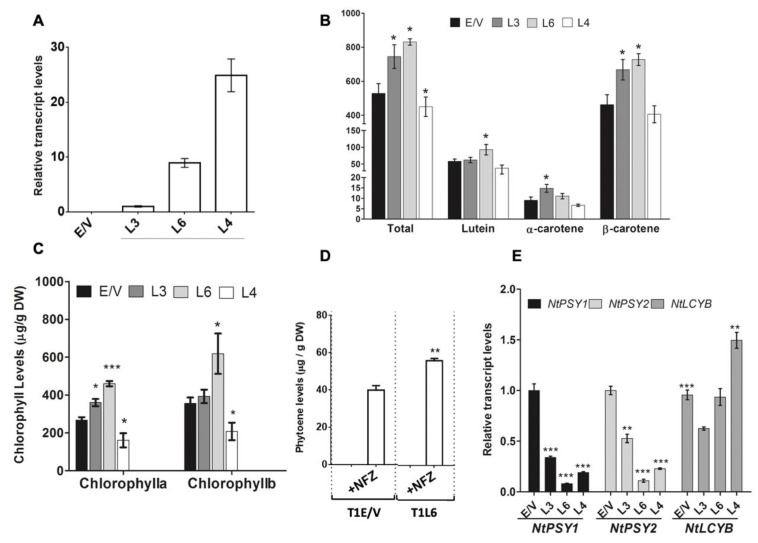
Carotenoid, phytoene and chlorophyll composition and expression of carotenogenic genes in *DcPSY2* transgenic *N. tabacum.* (**A**) Relative *DcPSY2* transcript expression in three independent T_0_ lines. Real-time RT-PCR was performed with tree biological replicates measured twice each. *rRNA18S* expression was used as normalizer and L3 was used as calibrator. (**B**) Total carotenoids, lutein, α-carotene and β-carotene composition and (**C**) chlorophyll a and chlorophyll b quantification in the three independent *N. tabacum* lines that express *DcPSY2*. Measurements were conducted by spectrophotometry and HPLC-RP in leaves of two-month-old T_0_ tobacco transgenic lines. E/V: Empty vector transgenic lines (control). Pigments were identified according to the retention time and absorption spectra. (**D**) Phytoene was quantified by HPLC-RP at 285 nm in three pools of three T_1_E/V and T_1_L6 lines each, after 100 µM Norflurazon treatment. (**E**) Relative expression levels of endogenous carotenogenic genes in *DcPSY2* T_0_ tobacco transgenic lines. Real-time RT-PCR was performed with tree biological replicates measured twice each for *NtPSY1*, *NtPSY2* and *NtLCYB*. Ubiquitin was used as housekeeping gene and E/V samples were used as calibrator and settled in 1. Error bars represent standard error of the mean (two technical replicates; asterisks indicate Student’s *t*-test significant differences (* *p* < 0.05, ** *p* < 0.01, *** *p* < 0.001).

**Figure 4 plants-12-01925-f004:**
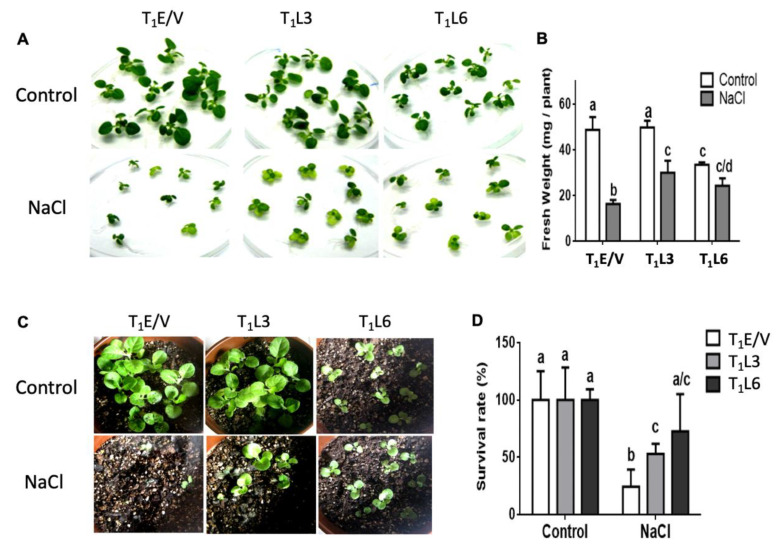
Salt stress tolerance in *DcPSY2* transgenic tobacco plants. (**A**) Phenotype of representative T_1_E/V, T_1_L3 and T_1_L6 cultivated in MS medium for 16 days in absence (control) or presence (NaCl) of 250 mM NaCl. (**B**) Fresh weight of the seedlings of T_1_E/V, T_1_L3 and T_1_L6 transgenic lines showed in (**A**) after 16 days grown in absence or presence of 250 mM NaCl. Error bars represent standard error of the mean (three biological replicates). Letters denote significant differences by two-tailed unpaired Student’s *t*-test (*p* < 0.05). (**C**) Plants from (**A**) were transplanted to greenhouse and watered for 21 days. The phenotype of T_1_E/V, T_1_L3, and T_1_L6 plants after 21 days of recovery in the greenhouse is shown. (**D**) Survival rate of representative T_1_E/V, T_1_L3 and T_1_L6 plants shown in (**C**) after 21 days of recovery in the greenhouse. Error bars represent standard error of the mean (three biological replicates). Different letters denote significant differences by two-tailed unpaired Student’s *t*-test (*p* < 0.05).

**Figure 5 plants-12-01925-f005:**
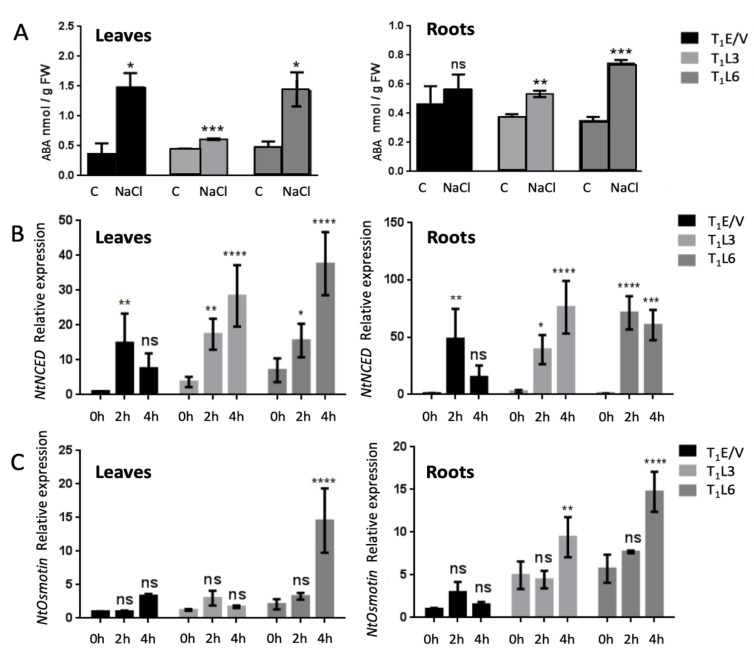
Abscisic acid quantification and relative expression of *NtNCED* and *NtOsmotin* in leaves and roots of transgenic tobacco lines subjected to acute salt stress. (**A**) ABA level quantified by HPLC-ESI-MS/MS in leaves and roots of a pool of 3 representative one-month-old T_1_E/V, T_1_L3 and T_1_L6 lines after 0 h (**C**), and 8 h (NaCl) of 250 mM NaCl treatment. Three biological replicates were performed. (**B**) *NtNCED3* and (**C**) *NtOsmotin* relative transcript abundance measured in leaves and roots after 0, 2 and 4 h of 250 mM NaCl treatment. Three biological replicates, each composed of three plants, were performed and real-time RT-PCR was measured in triplicate. *EF1 alpha* was used as housekeeping gene and each E/V 0h (control) was used as calibrator and settled in 1. Error bars represent standard error of the mean. Asterisks denote significant differences by two-tailed unpaired Student’s *t*-test (* *p* < 0.05, ** *p* < 0.01, *** *p* < 0.001, **** *p* < 0.001). ns denote no significant differences.

**Figure 6 plants-12-01925-f006:**
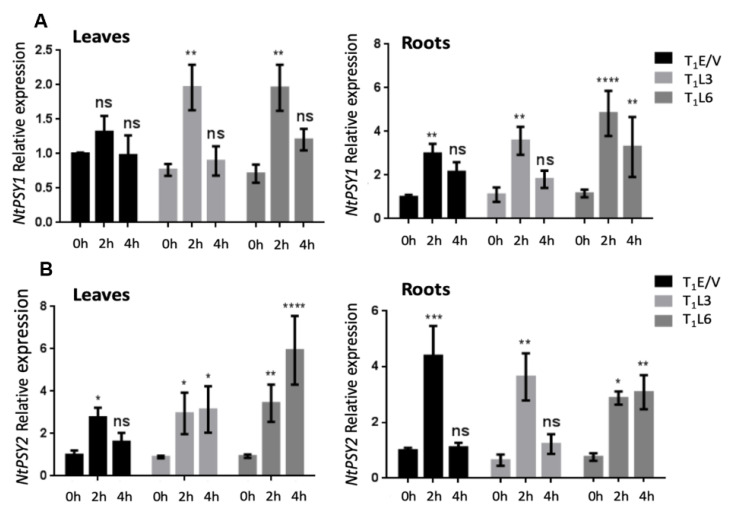
Endogenous *NtPSY1* and *NtPSY2* expression in tobacco *DcPSY2* transgenic lines after acute salt stress treatment. (**A**) *NtPSY1* and (**B**) *NtPSY2* relative transcript abundance measured in leaves and roots of T_1_E/V, T_1_L3 and T_1_L6 after 0 h, 2 h and 4 h of 250 mM NaCl treatment. Three biological replicates, each composed of three T_1_ plants, were performed, and real-time RT-PCR was measured in triplicate. *EF1 alpha* expression was used as normalizer and each E/V 0 h (control) was used as calibrator and settled in 1. Error bars represent standard error of the mean. Asterisks denote significant differences by two-tailed unpaired Student’s *t*-test (* *p* < 0.05, ** *p* < 0.01, *** *p* < 0.001, **** *p* < 0.001). ns denote no significant differences.

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
