# Peer review of "The Carrot Phytoene Synthase 2 (DcPSY2) Promotes Salt Stress Tolerance through a Positive Regulation of Abscisic Acid and Abiotic-Related Genes in Nicotiana tabacum"

_plants, 2023, doi:10.3390/plants12101925_

Round 1

Reviewer 1 Report

This is an interesting, but not fully completed work. Significant revision of the text is required. English language must also be checked throughout the text. The most important thing that authors should pay attention to is the Abstract and the Conclusion sections. The main question is why plants of the T0 generation were used in real-time experiments. In addition, the presence of DcPSY2 gene reduced the growth of tobacco plants, but increased their resistance to salt stress. Therefore, it is impossible to speak unambiguously about the benefits of obtaining such plants.

Title

From the title it should be clear what protein the abbreviation DcPSY2 corresponds to.

Abstract

The Abstract is written vaguely. It should be rewritten.

“In higher plants, carotenoid biosynthesis begins with the generation of phytoene from two molecules of geranylgeranyl pyrophosphate, a process catalyzed by phytoene synthase (PSY), a key regulatory step in the pathway.” Should be rephrased.

“Here we show that the ectopic expression of DcPSY2 in Nicotiana tabacum L. (tobacco) produces a significant increase in phytoene levels, total carotenoids, and chlorophylls specially in those lines with the lowest expression level.” What lines are you talking about? Why is the level of synthesis in them lowered?

“Tobacco lines subjected to chronic saline stress shows between 2- and 3,6-fold increase in survival rate regarding to control lines,…” Did you mean transgenic lines of tobacco?

“Conclusions: These results present evidences on the functionality of DcPSY2 in conferring salt stress tolerance in transgenic lines.” What lines are you talking about?

Materials and Methods

All information about obtaining generations T0, T1 and what they were used for should be further described in the section “Plant Material and Agrobacterium mediated transformation”. The methodological details of obtaining generation T1 plants from the Results should be transferred to Methods section.

About real-time PCR. How was the quality of the RNA checked? Were the lengths and purity of the amplified fragments checked by electrophoresis?

Results

Why were not T1 plants more stably expressing the introduced gene used in all real-time PCR experiments? Why were different plant generations used in different experiments? What was the reason for the choice in each case?

T1 generation plants were not analyzed separately for DcPSY2expression and were used as a pool? Why?  Such pool of plants cannot have the same name as the plant of the T0 generation. This introduces confusion in understanding which plants are talking about now.

“N. tabacum lines expressing moderate DcPSY2 level shows higher carotenoid content” What does moderate level mean?

What did mature transgenic plants look like?

Figure 4С. Missing signature control, NaCl

Discussion

“Total carotenoid and chlorophyll a content correlated inversely with the relative tran-script abundance of DcPSY2 transgene in each line. Specifically, lower and moderate (L3 and L6) expression levels of DcPSY2 lead to significant increases in carotenoid and chlorophyll content, whereas strong expression (L4) results in unaltered or reduced levels (Figure 3A, B, C).” Transgenic lines are compared according to the level of gene expression with each other. Therefore, expressions “the highest, the lowest, and moderate expression levels” should be used.

“This phenomenon has been observed in other studies and species (Busch et al., 2002, Maass et al., 2009), demonstrating the intimate association between PSY expression and the regulation of carotenoid accumulation. As an explanation for this phenotype, we hypothesize that low and moderate DcPSY2 expression leads to an increased carotenoid content through a retrograde-crosstalk mechanism with chlorophyll biosynthesis (Moreno et al., 2016) and an increment of transgene abundance over a threshold level induces an opposite effect on carotenoid accumulation.” Maybe this is due to the fact that the plants of generation T0 are characterized by mosaicity. Additionally, inserting the foreign gene into plant genome is always random and it may also affect other genes that results in gene silencing or altered expression.

Conclusion

Сonclusion is missing. It should display the main results of the work.

Taking into account the downregulation of tobacco's own genes NtPSY1 and NtPSY2 and the much smaller size of transgenic plants, in my opinion it cannot be argued that “Transgenic DcPSY2 tobacco plants presenting salt stress tolerance..are promising approach for economically important crops.”

Author Response

Title

Q1: From the title it should be clear what protein the abbreviation DcPSY2 corresponds to.

R1: Thanks for the suggestion; the name Phytoene synthase was included

Abstract

Q2: The Abstract is written vaguely. It should be rewritten.

R2: We rewrite the abstract following your suggestion and being more precise in some sentences.

Q3: “In higher plants, carotenoid biosynthesis begins with the generation of phytoene from two molecules of geranylgeranyl pyrophosphate, a process catalyzed by phytoene synthase (PSY), a key regulatory step in the pathway.” Should be rephrased.

R3: It was rephrased. I hope sounds better now.

Q4:“Here we show that the ectopic expression of DcPSY2 in Nicotiana tabacum L. (tobacco) produces a significant increase in phytoene levels, total carotenoids, and chlorophylls specially in those lines with the lowest expression level.” What lines are you talking about? Why is the level of synthesis in them lowered?

R4: We rephrased the sentence hoping is clearer. We are talking about L3 and L6, which presented lower level of DcPSY2 expression than L4, but presented a significant increase in total carotenoids, and chlorophylls, which is similar than reported by Busch et al 2002 by the overexpression of NtPSY

Q5:“Tobacco lines subjected to chronic saline stress shows between 2- and 3,6-fold increase in survival rate regarding to control lines,…” Did you mean transgenic lines of tobacco?

R5: Yes, we rewrite the sentence

Q6:“Conclusions: These results present evidences on the functionality of DcPSY2 in conferring salt stress tolerance in transgenic lines.” What lines are you talking about?

 R6: We refer to T1 lines derived from L3 and L6. We include more details in this new version.

Materials and Methods

Q7: All information about obtaining generations T0, T1 and what they were used for should be further described in the section “Plant Material and Agrobacterium mediated transformation”. The methodological details of obtaining generation T1 plants from the Results should be transferred to Methods section.

R7: The new version includes this change.

Q8: About real-time PCR. How was the quality of the RNA checked? Were the lengths and purity of the amplified fragments checked by electrophoresis?

R8:The quality of RNA was checked by electrophoresis and by RT-PCR amplification of 180bp fragment of RNAr18S (the same used for qRTPCR) and by the amplification of the complete cds of DcPSY2 (1.3 kb), showing the good quality of the cDNA obtained as it was suitable to amplify a long fragment. We include these results in Supplementary Figure 5 for more detail.

Results

Q9: Why were not T1 plants more stably expressing the introduced gene used in all real-time PCR experiments? Why were different plant generations used in different experiments? What was the reason for the choice in each case?

R9: The reason was to initial analyze the T0 transgenic plants to ensure that they express the transgene. Additionally, we include the quantification of carotenoids and chlorophylls, but the most important results are about the salt stress tolerance that T1 lines confer. We used different pools of T1 derived from L3 and L6 for each experiment, showing that there is a transversal response of T1 and not only in selected lines.

Q10: T1 generation plants were not analyzed separately for DcPSY2expression and were used as a pool? Why?  Such pool of plants cannot have the same name as the plant of the T0 generation. This introduces confusion in understanding which plants are talking about now.

R10: We used different pools of T1 derived from L3 and L6 for each experiment, showing that there is a transversal response of T1 and not only in selected lines. We rename T1 Lines hoping that their interpretation is clearer

Q11:“N. tabacum lines expressing moderate DcPSY2 level shows higher carotenoid content” What does moderate level mean?

R11: We rephrase this sentence hoping that their interpretation is clearer.

Q12: What did mature transgenic plants look like?

R12: Although T1L6 are smaller in size than T1L3 and T1E/V plants during the first month, even in the absence of salt, they reach a phenotype similar to T1L3 and T1EVs at the adulthood suggesting that their development time is only slower

Figure 4С. Missing signature control, NaCl

Thank you! we include the signatures

Discussion

Q13: “Total carotenoid and chlorophyll a content correlated inversely with the relative tran-script abundance of DcPSY2 transgene in each line. Specifically, lower and moderate (L3 and L6) expression levels of DcPSY2 lead to significant increases in carotenoid and chlorophyll content, whereas strong expression (L4) results in unaltered or reduced levels (Figure 3A, B, C).” Transgenic lines are compared according to the level of gene expression with each other. Therefore, expressions “the highest, the lowest, and moderate expression levels” should be used.

R13: We include the terms the highest, the lowest, and moderate

Q13: “This phenomenon has been observed in other studies and species (Busch et al., 2002, Maass et al., 2009), demonstrating the intimate association between PSY expression and the regulation of carotenoid accumulation. As an explanation for this phenotype, we hypothesize that low and moderate DcPSY2 expression leads to an increased carotenoid content through a retrograde-crosstalk mechanism with chlorophyll biosynthesis (Moreno et al., 2016) and an increment of transgene abundance over a threshold level induces an opposite effect on carotenoid accumulation.” Maybe this is due to the fact that the plants of generation T0 are characterized by mosaicity. Additionally, inserting the foreign gene into plant genome is always random and it may also affect other genes that results in gene silencing or altered expression.

R14: We include this idea in this section.

Conclusion

Q15: Сonclusion is missing. It should display the main results of the work.

R15: We hope that we agree that the main conclusion of the work is that Transgenic DcPSY2 tobacco plants present salt stress tolerance with an increase in ABA level and in the expression of genes related to abiotic stress response

Q16: Taking into account the downregulation of tobacco's own genes NtPSY1 and NtPSY2 and the much smaller size of transgenic plants, in my opinion it cannot be argued that “Transgenic DcPSY2 tobacco plants presenting salt stress tolerance..are promising approach for economically important crops.”

R16: We refocus on rootstocks, for which it would not affect the size of the plant and they are also in direct contact with saline stress in the soil.

Reviewer 2 Report

This manuscript presents an investigation into carrot PSY genes and their possible role in salt tolerance. Overall, the data are presented well although I have some comments I like to see addressed before publication.  

 Comments:

 P 6        The authors state “The predicted proteins DcPSY1 (398 aa) and DcPSY2 (438 aa) share 86% identity and present the same amino acids at the active site, substrate binding pocket, Mg2+ binding site and aspartate rich region (Supplementary Figure 3) that are located in the squalene/phytoene synthase, isoprenoid synthase, and trans-isoprenyl diphosphate synthase domains found in functional PSY enzymes (Figure 2A).”

I suggest to use a program such as clustal omega to obtain indications for identical amino acids (**), and to identify the different domains in the same way for Supplementary Figures 3 and 4.  This will make it easier to follow the text. Note that it is difficult at the moment to distinguish between the dark and blue highlights in Figure S3.   Also, there seems to be a discrepancy between the 2 figures. For example, Figure S3 has an aspartate rich region (yellow highlight) of only 1 amino acid, whereas Figure S4 has two sites.

 P6 Be consistent in the abbreviation used: “EV plants” OR “E/V lines” and define (write name in full) the first time.

 P7 The fresh weight of L6 plants is lower than that of L3 plants according to figure 4B, and suggested by the pictures shown in figure 4C. So even though the survival rate of L6 plants is better, this does not lead to better growth, which means that L3 plants might have a higher yield also when the plants are older. This can be seen in Figure 4C. In light of this I suggest rephrasing the text in the CONCLUSIONS section (P11).

Supplementary figure 2. Would be useful to indicate the start and stop codons (by making those letter bold for example).

Figure 2:

REPLACE “rots” WITH “roots”

The text starting with “ubiquitin was used …”  is less clear. Please rewrite.

Asterisks are mentioned in legend but not present in figure 2B.

Rearrange text in panel A so that 410 is on one line

I suggest presenting expression data for DcPSY1 and DcPSY2 in the same order in Fig 2A and B

Figure 5: 5A: I suggest to write “ABA nmol/gFW” on x-axes

Author Response

P 6   The authors state “The predicted proteins DcPSY1 (398 aa) and DcPSY2 (438 aa) share 86% identity and present the same amino acids at the active site, substrate binding pocket, Mg2+ binding site and aspartate rich region (Supplementary Figure 3) that are located in the squalene/phytoene synthase, isoprenoid synthase, and trans-isoprenyl diphosphate synthase domains found in functional PSY enzymes (Figure 2A).”

I suggest to use a program such as clustal omega to obtain indications for identical amino acids (**), and to identify the different domains in the same way for Supplementary Figures 3 and 4.  This will make it easier to follow the text. Note that it is difficult at the moment to distinguish between the dark and blue highlights in Figure S3.   Also, there seems to be a discrepancy between the 2 figures. For example, Figure S3 has an aspartate rich region (yellow highlight) of only 1 amino acid, whereas Figure S4 has two sites.

R1: We re analyze and made a new Figure S3 hoping it is clearer

Q2: P6 Be consistent in the abbreviation used: “EV plants” OR “E/V lines” and define (write name in full) the first time.

R2: Thank you to notice it, we change to E/V

 Q3:P7 The fresh weight of L6 plants is lower than that of L3 plants according to figure 4B, and suggested by the pictures shown in figure 4C. So even though the survival rate of L6 plants is better, this does not lead to better growth, which means that L3 plants might have a higher yield also when the plants are older. This can be seen in Figure 4C. In light of this I suggest rephrasing the text in the CONCLUSIONS section (P11).

R3: Thank you to notice it, we include this information in the results and discussion sections

Q4: Supplementary figure 2. Would be useful to indicate the start and stop codons (by making those letter bold for example).

R4:The start and stop codons were highlighted in Sup Figure 1 because Figure 2 shows the alignment with respect to NtPSY2 that started for 184 bp.

Q5: Figure 2: REPLACE “rots” WITH “roots”; The text starting with “ubiquitin was used …”  is less clear. Please rewrite; Asterisks are mentioned in legend but not present in figure 2B; Rearrange text in panel A so that 410 is on one line; I suggest presenting expression data for DcPSY1 and DcPSY2 in the same order in Fig 2A and B

R5: Thank you for the comments, we change Figure 2B hoping it will be clearer to analyze the results

Q5: Figure 5: 5A: I suggest to write “ABA nmol/gFW” on x-axes

R6: Thank you, we include ABA nmol/gFW in y axes and we change also the order of x axes to keep the same format as the other figures. We hope all changes are to the liking of the reviewer.

Round 2

Reviewer 1 Report

The authors did a great job on the article. The article can be published after making minor corrections.

Abstract

“Here we show that the ectopic expression of DcPSY2 in Nicotiana tabacum L. (tobacco) produces 23 in L3 and L6 a significant increase in total carotenoids, and chlorophyll a, and a significant 24 increment in phytoene in T1L6.” Please, add the term “lines”

“Conclusions: These results present evidences on the functionality of DcPSY2 in conferring salt stress tolerance in transgenic T1L3 and T1L6 lines.” ….salt stress tolerance in transgenic tobacco lines (more common)

Results

“N. tabacum lines expressing DcPSY2 level show and increment in carotenoid content” The title of the section is unclear.

Figure 6 legend. A) is missing

“CONCLUSIONS” Conclusion? Lines 540-546 can be moved to this section as a conclusion.

Author Response

Dear reviewer
I really appreciate the thoroughness of your review which allowed us to improve the manuscript. I send the last revisions
requested (highlighted in yellow). I hope I have properly interpreted the requests.
Best regards,
Claudia Stange